# Random access quantum information processors using multimode circuit quantum electrodynamics

R.K. Naik [1], N. Leung[1], S. Chakram[1], Peter Groszkowski[2], Y. Lu[1], N. Earnest[1], D.C. McKay[3], Jens Koch [2] & D.I. Schuster[1]

Qubit connectivity is an important property of a quantum processor, with an ideal processor having random access—the ability of arbitrary qubit pairs to interact directly. This a challenge with superconducting circuits, as state-of-the-art architectures rely on only nearest-neighbor coupling. Here, we implement a random access superconducting quantum information processor, demonstrating universal operations on a nine-qubit memory, with a Josephson junction transmon circuit serving as the central processor. The quantum memory uses the eigenmodes of a linear array of coupled superconducting resonators. We selectively stimulate vacuum Rabi oscillations between the transmon and individual eigenmodes through parametric flux modulation of the transmon frequency. Utilizing these oscillations, we perform a universal set of quantum gates on 38 arbitrary pairs of modes and prepare multimode entangled states, all using only two control lines. We thus achieve hardware-efficient random access multi-qubit control in an architecture compatible with long-lived microwave cavity-based quantum memories.

[1] The James Franck Institute and Department of Physics, University of Chicago, Chicago, IL 60637, USA. [2] Department of Physics and Astronomy, Northwestern University, Evanston, IL 60208, USA. [3] IBM T.J. Watson Research Center, Yorktown Heights, NY 10598, USA. R.K. Naik, N. Leung and S. Chakram contributed equally to this work. Correspondence and requests for materials should be addressed to R.K.N. (email: rnaik24@uchicago.edu) or to D.I.S. (email: david.schuster@uchicago.edu)

Superconducting circuit quantum electrodynamics (cQED) is rapidly progressing toward small and medium-scale quantum computation[1]. Superconducting circuits consisting of lattices of Josephson junction qubits[2,3] have been used to realize quantum information processors relying on nearest-neighbor interactions for entanglement. An outstanding challenge in cQED is the realization of architectures with high qubit connectivity, the advantages of which have been demonstrated in ion trap quantum computers[4–6]. Classical computation architectures typically address this challenge by using a central processor that can randomly access a large memory, with the two elements often comprising distinct physical systems. We implement a quantum analog of this architecture, realizing a random access quantum information processor using cQED.

As in the classical case, quantum logic elements, such as superconducting qubits, are expensive in terms of control resources and have limited coherence times. Quantum memories based on harmonic oscillators, instead, can have coherence times two orders of magnitude longer than the best qubits[7–9], but are incapable of logic operations on their own. This observation suggests supporting each logic-capable processor qubit with many memory qubits. In the near term, this architecture provides a means of controlling tens of highly coherent qubits with minimal cryogenic and electronic-control overhead. To build larger systems compatible with existing quantum error correction architectures[10–13], one can connect individual modules consisting of a single processor qubit and a number of bits of memory while still accessing each module in parallel.

Here, we describe and experimentally demonstrate the use of a single non-linear element to enable universal quantum logic with random access on a collection of harmonic oscillators. We store information in distributed, readily accessible, and spectrally distinct resonator modes. We show how to perform single-qubit gates on arbitrary modes by using frequency-selective parametric control[14–19] to exchange information between a superconducting transmon qubit[20] and individual resonator modes. Next, using higher levels of the transmon, we realize controlled-phase (CZ) and controlled-NOT (CX) gates on arbitrary pairs of modes. Therefore, we demonstrate all the ingredients necessary for universal quantum computation with harmonic modes. Finally, we use these tools to prepare multi-mode entangled states as an important step toward quantum error correction.

## Results

**Multimode quantum memory**. To build a multimode quantum memory we use the eigenmodes of a linear array of $n = 11$ identical, strongly coupled superconducting resonators[21] (see Fig. 1). For a linear array, the eigenmodes correspond to distributed "momentum" states (see Supplementary Note 3). Importantly, every mode has non-zero amplitude at the edge, allowing the transmon to couple to each mode. The Hamiltonian of the combined system is:

$$\hat{H} = h\nu_q(t)\hat{a}^\dagger \hat{a} + \frac{1}{2}h\alpha\,\hat{a}^\dagger\hat{a}\big(\hat{a}^\dagger\hat{a} - 1\big) + \sum_{k=1}^{n} h\nu_k \hat{b}_k^\dagger \hat{b}_k$$
$$+ \sum_{k=1}^{n} hg_k\big(\hat{b}_k + \hat{b}_k^\dagger\big)\big(\hat{a} + \hat{a}^\dagger\big),$$

(1)

where the transmon is treated as a Duffing oscillator[20] with anharmonicity $\alpha$, coupled to the modes with frequency $\nu_k$ (6–7 GHz) and coupling strength $g_k$ (50–200 MHz,

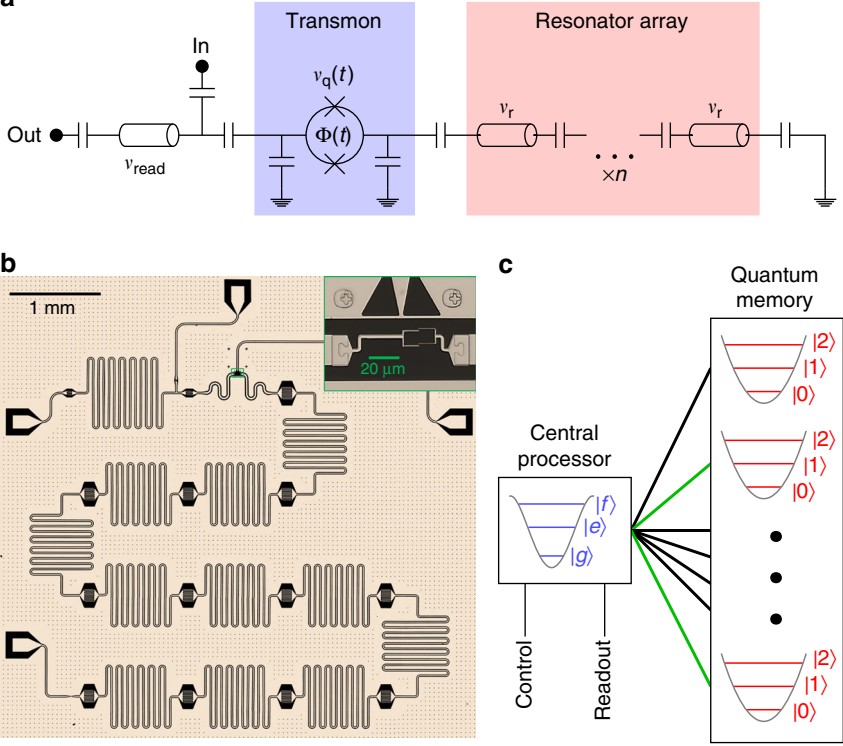

**Fig. 1** Random access superconducting quantum information processor. **a**, **b** Schematic and optical image, respectively, of the superconducting microwave circuit. The circuit comprises an array of 11 identically designed, co-planar waveguide (CPW) half-wave resonators, capacitively coupled strongly to each other. The top end of the array is capacitively coupled to a tunable transmon qubit. The transmon is measured with a separate resonator, whose input line doubles as a charge bias for the transmon. The inset shows the tunable SQuID of the transmon, as well as its flux bias above it. **c** Random access with multiplexed control. The quantum memory consists of the eigenmodes of the array, with each mode accessible to the transmon. This allows for quantum operations between two arbitrary memory modes (such as those highlighted in green) via the central processing transmon and its control lines

Supplementary Note 8). The operators $\hat{a}^\dagger$ ($\hat{a}$) and $\hat{b}_k^\dagger$ ($\hat{b}_k$) create (annihilate) photons in the transmon and in eigenmode $k$, respectively. While this implementation is straightforward, the

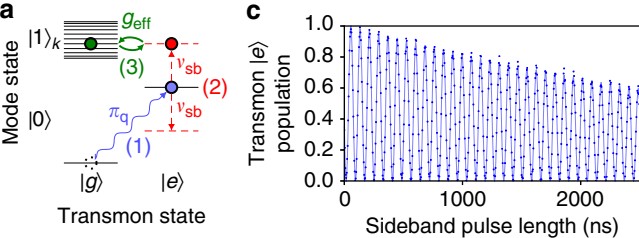

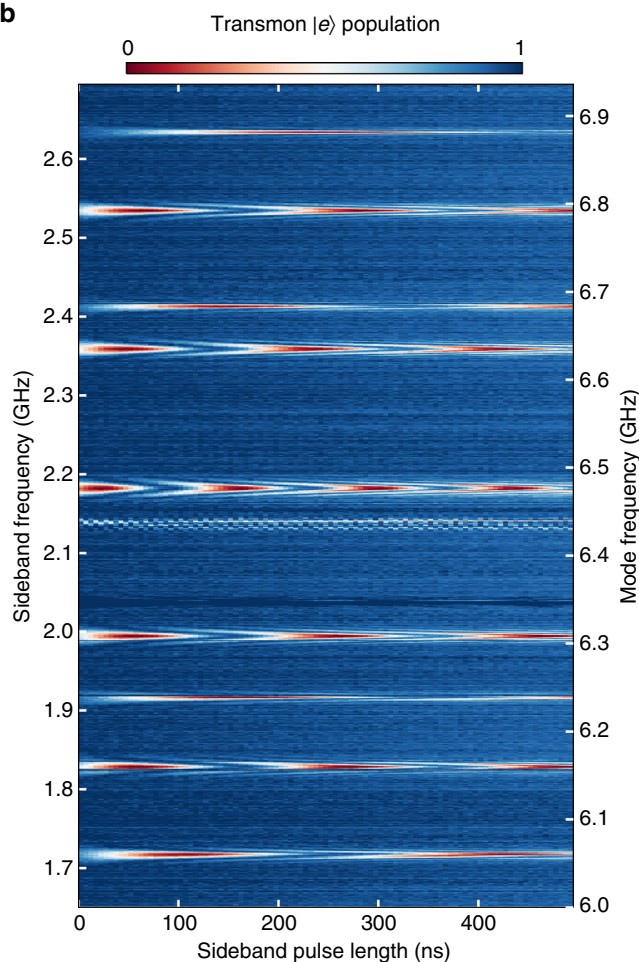

**Fig. 2** Stimulated vacuum Rabi oscillations. **a** Generation of stimulated vacuum Rabi oscillations. $|1\rangle_k$ is the state with a single photon in mode $k$; all other modes are in the ground state. (1) An excitation is loaded into the transmon via its charge bias. (2) The transmon frequency is flux-modulated to create sidebands. (3) When a sideband is resonant with a mode, single-photon vacuum Rabi oscillations occur between transmon and the mode. **b** Experimental results obtained from this protocol for a range of sideband modulation frequencies, with the transmon biased at $\nu_q = 4.28$ GHz. The length of the flux modulation pulse is swept for each frequency and the excited state population of the transmon is measured after the pulse ends. Chevron patterns indicate parametrically induced resonant oscillations with each of the memory modes. Two of the eleven modes are weakly coupled to the transmon and are not visible at these flux modulation amplitudes. The distribution of the modes can be understood through Hamiltonian tomography[48] (Supplementary Note 9). **c** Resonant oscillations between transmon and mode 6

idea of a multimode memory also applies to related systems with many harmonic degrees of freedom, including long transmission-line[22] or 3D waveguide cavities. We limit ourselves to the zero-photon and one-photon Fock states of the eigenmodes. It is also possible to use more of the oscillator Hilbert space, allowing logical encoding in terms of cat[23] and binomial code[24] states.

Given access to the multimode memory via the transmon, we demonstrate methods to address each mode individually. In many circuit QED schemes, excitations are loaded into modes by adiabatically tuning the qubit frequency through or near a mode resonance[25]. This works well for single modes, but for a multimode manifold one must carefully manage Landau–Zener transitions through several modes[21], to avoid leaving residual excitations elsewhere in the manifold. Also, the qubit must be returned to the far-dispersive regime to minimize spurious unwanted interactions, requiring longer gate durations.

**Stimulated vacuum Rabi oscillations**. We induce resonant interactions between the transmon and an individual mode by modulating the transmon excitation energy via its flux bias. The modulation creates sidebands of the transmon excited state, detuned from the original resonance by the frequency of the applied flux tone. When one of these sidebands is resonant with a mode of the memory, the system experiences stimulated vacuum Rabi oscillations: parametrically induced exchange of a single photon between the transmon and the selected mode. These are similar to resonant vacuum Rabi oscillations[26], but occur at a rate that is controlled by the modulation amplitude[14,15] $g_{\text{eff},k} = g_k J_1(\epsilon / 2\nu_{\text{sb}})$, where $J_1$ is the first Bessel function, $\epsilon$ and $\nu_{\text{sb}}$ are the amplitude and frequency of the modulation, respectively, and $g_k$ is the bare coupling rate to eigenmode $k$ (Supplementary Note 4). The rate of photon exchange is linear to lowest order in $\epsilon$ and can be as large as $g_k/2$.

To illustrate the application of parametric control for addressing the multimode memory, we employ the experimental sequence shown in Fig. 2a. First, the transmon is excited via its charge bias. Subsequently, we modulate the flux to create sidebands of the transmon excited state at the modulation frequency. This is repeated for different flux pulse durations and frequencies, with the population of the transmon excited state measured at the end of each sequence. When the frequency matches the detuning between the transmon and a given eigenmode, we observe full-contrast stimulated vacuum Rabi oscillations. In Fig. 2b, we see the resulting characteristic chevron patterns[15] as the modulation frequency approaches the detuning between the transmon and each of the modes. For long modulation times, the excited state population approaches zero. This is evident in the stimulated vacuum Rabi oscillation between the transmon and mode 6 shown in Fig. 2c. This indicates that the original photon is being exchanged between the transmon and the mode and no other photons are being pumped into the system. We achieve photon exchange between the transmon and individual modes in 20–100 ns, depending on the mode. This rate is limited by spectral crowding arising from neighboring modes and sideband transitions involving the transmon $|f\rangle$ level. This operation is coherent and can be used to transfer arbitrary qubit states between the transmon and the memory mode, corresponding to a transmon-mode iSWAP[27] in the single-excitation subspace.

**Universal quantum control**. The transmon-mode iSWAP and arbitrary rotations of the transmon state via its charge bias provide a toolbox for universal state preparation, manipulation, and measurement of each mode of the quantum memory. In Fig. 3, we illustrate how to perform these operations. To characterize the

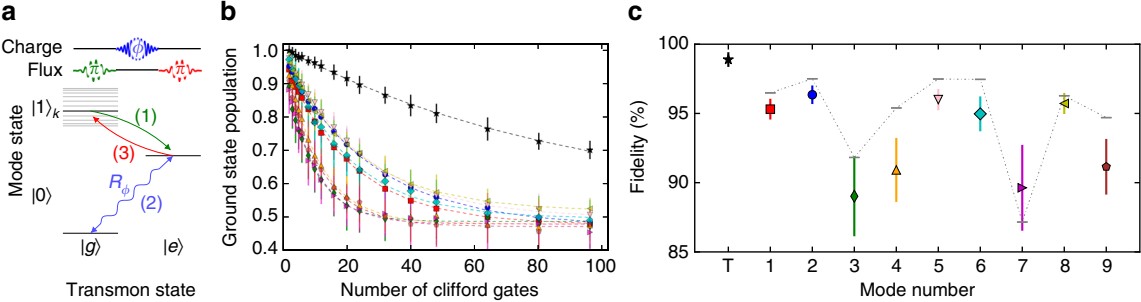

**Fig. 3** Single-mode gate protocol and benchmarking. **a** The sequence for generating arbitrary single-qubit gates of a memory mode: (1) The mode's initial state, consisting of a superposition of 0 and 1 photon Fock states, is swapped to the transmon (initially in its ground state), using a transmon-mode iSWAP (see text). (2) The transmon is rotated by the desired amount ($R_\phi$) via its charge control line. (3) The rotated state is swapped back to the mode, by reversing the iSWAP gate in (1). Segments of this sequence are used to achieve state preparation [steps (2) and (3)] and measurement [steps (1) and (2)] of each mode. **b** Single-mode RB. We apply sequences of varying numbers of consecutive Clifford gates, then invert each sequence with a unique Clifford gate. We measure the transmon ground-state population after inversion and average over 32 different random sequences, with the standard deviation (s.d.) plotted as error bars for each sequence length. **c** From fitting the resulting data, we find single-mode gate fidelities from $89.0 \pm 2.9$ to $96.3 \pm 0.7\%$ and a transmon ($T$ in the figure) gate fidelity of $98.9 \pm 1.3\%$. These are consistent with the expected coherence-limited fidelities, plotted as gray bars (s.d. from fit plotted as error bars)

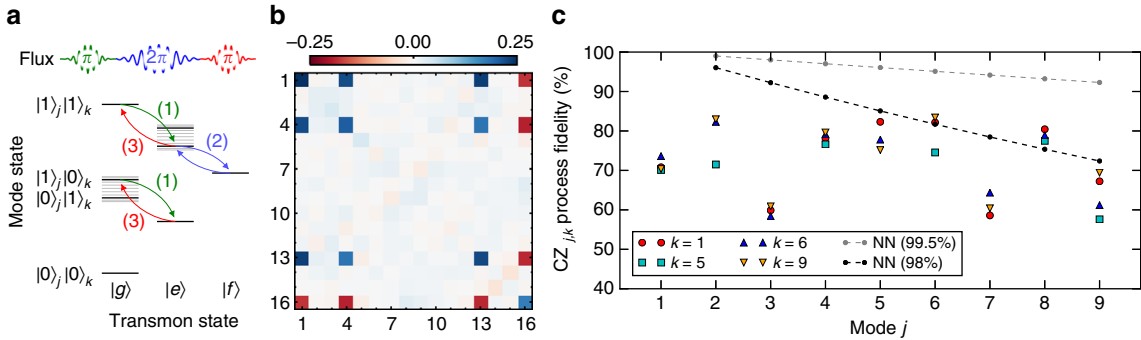

**Fig. 4** Controlled-phase gate between two arbitrary modes. **a** Protocol for controlled-phase (CZ) gate between an arbitrary pair of modes, with $j$ indicating the control mode and $k$ indicating the target mode of the gate: (1) The state of mode $j$ is swapped to the transmon via a transmon-mode iSWAP pulse at the frequency difference between the transmon $|g\rangle - |e\rangle$ transition and mode $k$. (2) A CZ gate is performed between mode $k$ and the transmon, by applying two frequency-selective iSWAPs from energy level $|e1\rangle$ to level $|f0\rangle$ and back, mapping the state $|e1\rangle$ to $-|e1\rangle$. (3) The state of the transmon is swapped back to mode $j$, reversing the iSWAP in (1). **b** Process matrix for the CZ gate between modes $j = 6$ and $k = 8$, corresponding to a process fidelity of 82% (see Supplementary Note 17 for details on state preparation and measurement). **c** Fidelities from process tomography for 38 pairs of memory modes with $k = 2, 5, 6, 8$. The process fidelities are extracted from sequences that include SPAM errors, and are conservative estimates of the gate fidelities. For comparison, the dashed black and gray lines show the decay in fidelity for a two-qubit gate between qubit 1 and qubit $j$ in a corresponding linear array comprising only nearest-neighbor gates with fidelities of 99.5[35] and 98%, respectively

quality of our single-mode operations, we perform randomized benchmarking (RB)[28,29]. We generate single-mode Clifford operations by sandwiching single-qubit Clifford rotations of the transmon with transmon-mode iSWAPs (Supplementary Note 12). We achieve RB fidelities ranging from $89.0 \pm 2.9\%$ to $96.3 \pm 0.7\%$. These fidelities approach the expected coherence limit, indicated by the gray bars in the figure. The coherence limits are estimated based on the qubit RB fidelity, the iSWAP times (20–100 ns), and the coherence times ($T_1 = 1$–5 µs, $T_2^* = 1$–8.5 µs) of individual modes (Supplementary Note 10). Each single-mode gate consists of two transmon-mode iSWAPs, and a single transmon gate. From the error in the single-mode and transmon RB, we estimate the fidelities of the individual transmon-mode iSWAP operations to range from 95 to 98.6%.

To achieve universal control of the quantum memory, we extend our parametric protocols to realize two-mode gates. We perform conditional operations between the transmon and individual modes by utilizing the $|e\rangle - |f\rangle$ transition of the transmon. A controlled-phase (CZ) gate between the transmon

and an individual mode consists of two sideband iSWAPs resonant to the $|e1\rangle - |f0\rangle$ transition, selectively mapping the state $|e1\rangle$ to $-|e1\rangle$, leaving all other states unchanged due to the anharmonicity of the transmon (Supplementary Note 6). To enact a CZ gate between two arbitrary modes, the control mode is swapped into the transmon, a transmon-mode CZ is performed, and the mode is swapped back as illustrated in Fig. 4a. In our device, gate speeds (250–400 ns) are primarily limited by crosstalk between iSWAP operations on the $|g\rangle - |e\rangle$ and $|e\rangle - |f\rangle$ transitions of modes with difference frequencies approaching the anharmonicity of the transmon. This crosstalk can be reduced by tailoring the frequency spacing of the memory modes and the anharmonicity of the transmon. In addition to the CZ gate, we obtain controlled X and Y gates (CX, CY) between modes by swapping $|e\rangle$ and $|f\rangle$ transmon state populations in the middle of the pulse sequence for the CZ gate. These gate protocols can be extended to realize two-mode SWAP gates (Supplementary Note 13), as well as multi-qubit gates such as Toffoli and controlled-controlled-phase (CCZ) gates[30] between arbitrary modes.

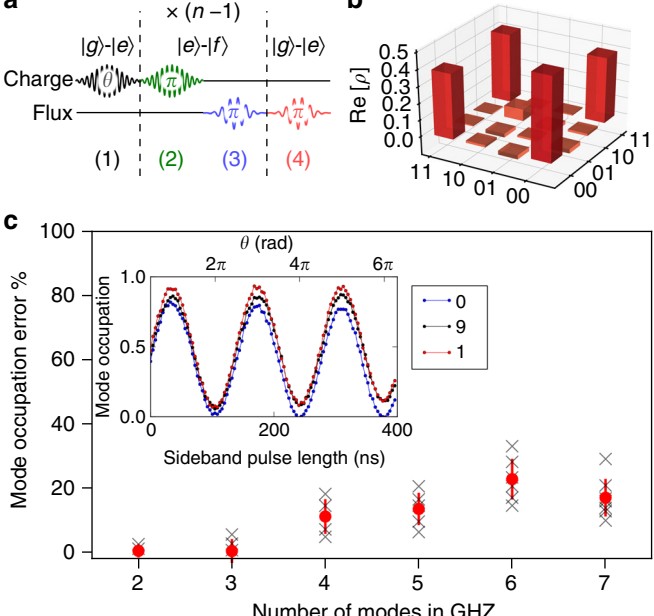

**Fig. 5** Multimode entanglement. **a** Pulse sequence for generating *n*-mode maximally entangled states. Step (1) creates a superposition of the transmon $|g\rangle$ and $|e\rangle$ states, with the relative amplitudes of the superposition controlled by the rotation angle $\theta$. Steps (2) and (3) load photons into modes of the memory, conditioned on the transmon state by utilizing the transmon $|f\rangle$ state. These steps are repeated $n-1$ times to entangle additional modes. Step (4) performs a $|g\rangle - |e\rangle$ iSWAP to the last mode, disentangling the transmon from the modes. **b** The real part of the density matrix $\langle\rho\rangle$ of the $|\Phi^{+}\rangle$ Bell state of mode 6 and 8, obtained from tomography. **c** (inset) Correlated oscillations resulting from sweeping $\theta$ and measuring each mode individually. This demonstrates control of the relative amplitudes of the entangled state superposition. **c** Deviation from expected mean populations of each of the modes, upon preparation of the GHZ state ($\theta = \frac{\pi}{2}$). The red filled circles and error bars indicate the average and s.d., respectively, over individual mode measurements (black crosses)

To perform high-fidelity gates between modes, several issues must be considered. These include: (1) DC shifts of the transmon frequency during iSWAP pulses (~10 MHz), (2) dispersive shift of the $|e1\rangle$ state (~1 MHz), and (3) stimulated dispersive shifts of non-targeted modes during iSWAP pulses (~10–100 kHz). We fully compensate effect (1) and correct the phase error arising from (2) by calibrating the phase errors and suitably adjusting the relative phases of the iSWAP pulses (Supplementary Note 14). The error from (3) is relatively small and currently adds to the gate error.

Our multimode architecture allows for straightforward measurements of arbitrary multi-bit correlators, forming a basis for tomography, and for the stabilizer measurements required for error correction. An arbitrary correlator comprises products of Pauli operators applied to each of the memory bits, and corresponds to a generalized parity measurement. This is exactly the back-action on the phase measurement of a transmon while serving as the control of a CZ (CX) gate targeting a memory mode[31]. The value of an arbitrary stabilizer can thus be measured by performing Ramsey interferometry of the transmon with a series of CZ (CX) gates applied to the desired memory modes.

We use correlator measurements to characterize a CZ gate between a given pair of modes via process tomography. We perform process tomography by applying the gate on 16 linearly independent input states that form a basis for an arbitrary two-qubit density matrix[32]. The resulting density matrices are reconstructed through state tomography. For two-qubit state tomography, we map all correlators to individual measurements of the transmon, using combinations of single-mode and two-mode gates.

In order to obtain a fair estimate of the gate fidelity, each of the process tomography sequences has a single two-mode gate. Additional gates required for tomography are combined with the characterized CZ gate (Supplementary Note 17). The process matrix obtained using this protocol for a CZ gate between modes 6 and 8 is shown in Fig. 4b. We use this protocol to characterize the fidelities for gates between 38 mode pairs, as shown in Fig. 4c. The fidelities from full process tomography range approximately from 60 to 80% for the CZ gates between the mode pairs indicated. These fidelities incorporate state preparation and measurement (SPAM) errors, with the SPAM sequences being of similar duration as the gates. Conservative estimates from single-mode and transmon RB (see Fig. 3c) give SPAM errors of 5–10%, depending on the modes addressed. The gate fidelities are largely limited by the coherence times of the modes (~5–15% error). Future devices based on 3D superconducting cavities[7] may have up to three orders of magnitude enhancement in memory mode coherence times. The process fidelities are additionally limited by dephasing of the transmon (~5% error), and residual coherent errors arising from bare and stimulated dispersive shifts. The error from the dephasing can be reduced by coupling a fixed-frequency transmon to the multimode memory using a tunable coupler[17,33,34]. Additionally, biasing the tunable coupler at a point with small static coupling also reduces coherent errors from the bare dispersive shift.

Figure 4c highlights the advantages of random access in a quantum computing architecture. An entangling gate between the first and the *j*th qubit of an array with only nearest-neighbor coupling would require $2j-1$ gates (such as CXs or iSWAPs). This results in an exponential decay of the fidelity with increasing distance between the corresponding qubits. Conversely, in a random access quantum information processor, there is no additional computational cost to perform gates between arbitrary pairs of qubits. Even without considering potential improvements in the coherence times, we see (Fig. 4c) that the processor performs competitively with state-of-the-art gates[35] between distant qubits in a nearest-neighbor architecture. While we have highlighted the advantages of this processor in terms of random access and minimal control hardware, a resulting requirement is the need to perform sequential operations. The number of modes which can be multiplexed to a single qubit without loss of fidelity is given by the ratio of the loss from idling in a cavity mode to the loss in performing qubit operations, which for modern 3D cavities can be up to 100[7].

**Multimode entanglement.** We use universal control of the quantum memory to build maximally entangled states spanning several modes, using the protocol described in Fig. 5a. First, we create a superposition of the transmon ground and excited states. Next, we add a photon to the desired mode, conditioned on the transmon state. This is repeated for each mode in the entangled state. Finally, we disentangle the transmon from the memory modes, transferring the remaining population into the final mode. In Fig. 5b, we show full state tomography for a Bell state[36] with state fidelity $F = 0.75$, including errors from tomography (Supplementary Note 16). In the inset of Fig. 5c, we apply the protocol to three modes and show populations of each of the modes as a function of the initial qubit rotation angle, $\theta$. Finally, in Fig. 5c, we show the population error from the target state at $\theta = \pi/2$, corresponding to a photonic Greenberger–Horne–Zeilinger (GHZ) state[37] of up to seven modes. While the three mode GHZ

state can be demonstrated to be tripartite entangled through a measurement of the Mermin witness[25,38] (Supplementary Note 19), full characterization of entangled states of more than two modes is hampered by the additional gates required for tomography and the gate fidelities of the current device. This protocol, however, illustrates the ease with which a random access quantum information processor can be used to generate multimode entangled states of arbitrary modes. Variants of this sequence can be used to create other classes of multimode entangled states, including W states, Dicke states[39], and cluster states[40]. Such states are valuable resources in several quantum error correction schemes and are useful for quantum optics and sensing[41].

With minimal control-hardware overhead, we perform universal quantum operations between arbitrary modes of a nine-qubit memory using a single transmon as the central processor. The methods described in this work extend beyond this particular implementation of a multimode memory and in particular are compatible with the use of 3D superconducting cavities, which are naturally multimodal and have demonstrated the longest coherence times currently available in cQED[7], with the potential for even further improvements[42]. This architecture is compatible with the error-correcting codes that use higher Fock states of a single oscillator, such as the cat[9,43] and binomial[24] codes, as well as distributed qubit codes[44,45], and is ideally suited to explore the potentially rich space of multi-qudit error-correcting codes that lie in between the two regimes[46,47]. This makes cQED-based random access quantum information processors a promising new module for quantum computation and simulation.

**Data availability**. The data that support the findings of this study are available on reasonable request from authors.

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

## Acknowledgements

The authors thank J. Simon, R. Ma, A. Oriani, R. Chakraborty, and A. Houck for useful discussions. The authors thank D. Czaplewski and P. Duda for support with device fabrication. This material is based on work supported by the Army Research Office under W911NF-15-1-0421. This work was partially supported by the University of Chicago

Materials Research Science and Engineering Center, which is funded by the National Science Foundation under award number DMR-1420709 and the Packard Foundation Fellowship for Science and Engineering. Use of the Center for Nanoscale Materials, an Office of Science user facility, was supported by the U.S. Department of Energy, Office of Science, Office of Basic Energy Sciences, under Contract No. DE-AC02-06CH11357. This work made use of the Pritzker Nanofabrication Facility of the Institute for Molecular Engineering at the University of Chicago, which receives support from SHyNE, a node of the National Science Foundation's National Nanotechnology Coordinated Infrastructure (NSF NNCI-1542205).

## Author contributions

N.L., R.K.N., and S.C. designed the device. R.K.N. and N.L. fabricated the device. S.C., R.K.N., and N.L. designed the experimental protocols, performed the experiments, and analyzed the data. P.G. and J.K. provided theoretical support. Y.L. and N.E. provided fabrication and experimental support. R.K.N., D.C.M., N.L., S.C., and D.I.S. conceived the experiment and all authors co-wrote the paper.

## Additional information

**Competing interests:** The authors declare no competing financial interests.

**Change History:** A correction to this article has been published and is linked from the HTML version of this paper.

