## [Peer Review File · Nature Communications]

Reviewers' comments:

Reviewer #1 (Remarks to the Author):

The authors of the manuscript "Random access quantum information processors" describe their experiments on a coupled multi-resonator transmon system. In particular they use the eigenmodes of the coupled resonators as a 9 qubit memory and show that the transmon allows to do arbitrary single memory-qubit operations. More impressively, this architecture also allows to do operations between arbitrary pairs of memory qubits. This is beautiful work which seems to be executed very carefully, as seen for example by the treatment of the calibration of all the relative phases. Both the manuscript and its supplementary information are well written. I have very few comments and without doubt this paper deserves to be published in Nature Communications with almost no modifications.

The few comments I can raise are the following:

- It is not obvious to me from the text why mode 2 and mode 8 have been excluded. I presume they don't couple or at most very weakly. It is a little confusing they are kept for example in fig. 3c, and I was searching for a while why there are 11 modes but only a 9-qubit memory is claimed.
- Perhaps the authors could mention as a downside of their approach that all operations will have to be performed serially, whereas in architectures with different connectivity some operations can usually be performed in parallel.
- It is unfortunate that the resonator T1 times are actually smaller than the transmon's T1. If the authors have an idea of the origin (the coupling capacitors?), it would be interesting to mention. The advantage of the resonators remains the increased T2*.
- I have a slight objection to label the normalized read-out signal as a transmon excitation probability between 0 and 1 (e.g. in Fig. 2bc), especially with a read-out fidelity between 0.3 and 0.85. I guess this also causes the error bars in Fig. 3b to go above 1, which is a little odd.

Although it does not affect the story at all, it would be worth a sentence or two in the main text.

Reviewer #2 (Remarks to the Author):

This work describes the implementation of a quantum processor comprising an 11-quantum-bit memory, realized as a coupled cavity array of superconducting resonators, and a single processing unit, a transmon-type structure, located at the edge of the array and yet coupled to all its eigenmodes. Coherent interaction between the transmon and a target eigenmode is enabled on-demand by parametric modulation of the transmon frequency. A modification of this scheme enables controlled interactions between any two eigenmodes, which, together with standard single-qubits gates, realize a universal set of pair operations within the quantum memory. The scheme also allows for the measurement of arbitrary multi-bit correlators with minimal overhead.

This is a solid, well-written, and carefully documented piece of work. Connectivity is *the* issue in current efforts aimed at quantum computing, and the multimode approach demonstrated here has some advantage over, say, nearest-neighbor interactions, which is evident in the scaling of two-qubit gate fidelities with the distance between the sites (on the other hand, a disadvantage of a single processing unit, if I understand it correctly, is that pairwise operations need to be implemented sequentially). While frequency crowding will likely limit the number of eigenmodes (and the speed of the gates), the modularity of the scheme makes it worth some consideration for near-term quantum computation and quantum simulation efforts (one could put, for instance, another transmon at the end of the array, and have these transmons shared between more arrays). Qubit lifetimes, resonator quality factors, and gate fidelities reported in this work are somewhat below the state of the art, but I believe that this is not due to an intrinsic limitation in the scheme and, therefore, should not diminish the importance of this work for the community. Before I make my final recommendation, I would like to ask the authors to carefully consider the technical comments below.

1) How do the coherence times of the eigenmodes compare to the quality factor of an individual resonator (fabricated and measured in a similar way)? Is there any relation between the spatial distribution of the modes and their coherence time? In the same way as you do "Hamiltonian reconstruction", could you do "loss reconstruction" and locate where the losses are in the array?

2) You find that T_2^* of the eigenmodes are not T_1 -limited, what is your explanation?

3) You have chosen to terminate the array with capacitors of the same size as the inter-site capacitors, so that (for otherwise identical resonators) the edge sites have the same bare frequency as any other and the Hamiltonian is of the form (1). That's fine (I would still mention it), but I find it misleading that in Fig. 1a you denote with g_r the capacitor that goes to ground. That capacitor does not correspond to any coupling element in the Hamiltonian, it just contributes to the total capacitance of the site. Furthermore (if I understand it correctly), the only reason why $g_q \neq g_r$ is that the transmon (as an anharmonic resonator) has a different impedance than the other sites. I would clarify these points and, in general, not associate g_r , g_q with the capacitors in Fig. 1a.

4) You find that modes 2 and 8 are much more weakly coupled than the others to the transmon (but still addressable). From "Hamiltonian reconstruction" you obtain weak tunnel coupling between sites 7 and 8 and 9-10, which you ascribe to two capacitors being "defective". Does it mean that their actual capacitance is much smaller than design value? But if it were so, should we not expect the bare frequencies of sites 7-10 to be all higher than the other sites? Please explain.

5) Page 4, last par., you say "Many of these errors can be reduced or eliminated by coupling a flux-insensitive transmon to the multimode memory". (i) "many of these errors" sounds too generic to me, can you just tell precisely which errors your suggestion is addressing? (ii) by "flux-insensitive" you mean, in this context, "frequency-fixed"?

6) Is it "impossible" in your scheme to enable multiple pair interaction at the same time? Or it can be done under certain conditions? I imagine that feature may find use in some applications.

7) In the Suppl Inf, Sec.V F, is the text accompanying Fig. 9 missing?

Reviewer #3 (Remarks to the Author):

In the manuscript Random access quantum information processors, the authors perform a proof-of-principle experiment towards an ideal processor showing random access, i.e. the capacity of the processor to directly access to every qubit in the quantum memory. They consider a superconducting transmon qubit playing the role of a central processor and the collective modes of an array of resonators as memory. The selective coupling between the processor and the eigenmodes of the array is performed via stimulated vacuum Rabi oscillations, which allows a faster state transfer than the usual resonant vacuum Rabi oscillations, which allow them to perform a universal set of single- and two-qubit gates and prepare highly entangled states, such as GHZ and Bell states. The authors claim that the approach is scalable and makes use only of the state-of-the-art technology.

In the battlefield which means achieving quantum supremacy in quantum technologies, and especially in superconducting circuits/circuit QED, there are roughly speaking two main approaches. The first one is led by Prof. Schoelkopf in Yale, among others, which privileges the large coherence times of resonators, using qubits as mere

non-linear elements to perform operations, and codify the information in cat states.

The second one, led by Prof. Martinis at UCSB/Google and Prof. DiCarlo in Delft, for instance, prefers to codify the information in superconducting qubits due to the possibility of direct manipulation, consigning resonators to perform measurements. In this context, the approach followed by the authors, although closer to the first approach, is original since it codifies the information in collective modes and looks for direct control of each memory qubit by the processor.

However, there are claims in this manuscript which are clearly oversold. For instance, they claim that their approach is scalable and it is not in a straightforward manner. Just by taking Eq. (3) of the Supplementary Material, one may see that the difference between consecutive eigenenergies is upper-bounded, in the worst case, by

$$\Delta E \leq 4gr$$

π

$n+1$. This means that, for the current case $n = 11$, they are already upperbounded by a separation of gr (in fact, following their data, it is even worse, since $\Delta E \approx 50 - 150\text{MHz}$, while $gr \approx 250\text{MHz}$). My impression is that they are close to the limit, since for larger n , the management of the Landau-Zener transitions should be cumbersome. A possible solution could be a kind of distributed memory in modules, each of them controlled by a single qubit, which can communicate. Indeed, the authors already foresee the problem and propose the aforementioned solution. However, as in the classical case, it will lead to problems such as von Neumann bottlenecks.

As a consequence of my previous objection, I am also skeptical about the statement that it is a promising route to achieve quantum supremacy. An optimistic estimation of the resources for the implementation of Grover's algorithm, the simplest useful quantum algorithm, with error correction makes use of the order of 500 million fullycontrollable qubits. Consequently, this claim is respectfully like stating that jumping is

a promising route to reach Alpha Centauri. Moreover, the experiment is interesting by itself and I do not think that it needs this kind of asserts.

I also do not agree with the easy claim that the methods described in this work are compatible with the use of 3D cavities. It is true that the coherent times of 3D cavities are around tens of miliseconds, but it does not directly mean that one may use the same approach, since the energy gaps between the collective modes could be smaller due to the degeneracy introduced by the orthogonal modes (see for instance, the works performed in the group of Prof. Gross at the WMI, Garching), which could destroy the information codification.

Even though the experiment is a proof of principle, quantum technologies and especially superconducting circuits / cQED are at the level of producing already applications beyond a set of universal quantum gates. Additionally, Nature Communications is an interdisciplinary journal, in which the articles should be accessible to non-specialist readers. In this sense, from my point of view, it is not only useful, but fundamental, to frame the new results in the state of the art. For instance, to my knowledge, the most advanced digital quantum simulation in this platform is achieved in R. Barends et al. Nature 534, 222 (2016). A fair comparison of the gates in terms of fidelities, times, robustness, etc is important, does not diminish the achievements of the authors, and could provide a more objective point of view to a non-specialist reader.

Additionally, the authors can find few more brief complementary comments about their manuscript.

Other Comments

1. I do not understand why the authors use the terminology memory bits and qubits alternatively. Sentences such as The memory bits are superpositions of vacuum and single-photon states in the abstract are conceptually wrong, since bit is a measurement of classical information. If the authors do not want to use the word qubit, the alternative is quantum bit. Otherwise, it seems that the memory is

classical and only able to store classical information.

2. Interdigitated capacitors is used only in the caption of Fig. (1). As this is not a broadly used terminology, I suggest them to replace it by capacitively coupled strongly to each other.

3. The points in Fig. (4c) are mainly indistinguishable. I suggest them to put the origin in a fidelity of 50%, which is essentially the worst possible fidelity for a quantum gate (corresponds to make it randomly), writing the legend in two columns, for instance.

4. With respect to the figures in the Supplementary Material, Fig. (5) has no legend in a., b. and d. Figures (7), (8), (9a), (10b) and (13) have no error bars.

2

5. As far as I see from Section IX-B of the Supplementary, the fidelity of Bell states is around 75%, which is compatible with the fidelity of the SWAP gate. However, the authors claim that they can construct GHZ states by repeating the swapping multiple times. With $n = 11$, it means the same number of entangling gates.

As a consequence, I would expect that the final GHZ state would show a terrible fidelity. It seems compatible with the fact that in Fig. (5c) mode occupation error percentage is provided, but no the fidelity. Authors must provide this value and, if it is too bad, they should avoid the claim that they can construct this state.

6. Finally, in References [9], [17] and [19] of the Supplementary Material, some capital letters are missing.

The authors of the manuscript “Random access quantum information processors” describe their experiments on a coupled multi-resonator transmon system. In particular they use the eigenmodes of the coupled resonators as a 9 qubit memory and show that the transmon allows to do arbitrary single memory-qubit operations. More impressively, this architecture also allows to do operations between arbitrary pairs of memory qubits. This is beautiful work which seems to be executed very carefully, as seen for example by the treatment of the calibration of all the relative phases. Both the manuscript and its supplementary information are well written. I have very few comments and without doubt this paper deserves to be published in Nature Communications with almost no modifications.

The few comments I can raise are the following:

- It is not obvious to me from the text why mode 2 and mode 8 have been excluded. I presume they don’t couple or at most very weakly. It is a little confusing they are kept for example in fig. 3c, and I was searching for a while why there are 11 modes but only a 9-qubit memory is claimed.

As the referee surmises, the two modes are excluded due to weak coupling to the transmon. The locations of the two modes are indicated by dotted lines in Supplementary Figure 3c. We determine the reason for this weak coupling to be defects in two of couplers in the resonator, as described in the section on Hamiltonian tomography (Section V. E.) in the supplementary information. We agree that it is confusing to include the modes in Figure 3 and 4 of the main text and have therefore removed them from these graphs, changing the numbering of the usable modes from 1 through 9.

- Perhaps the authors could mention as a downside of their approach that all operations will have to be performed serially, whereas in architectures with different connectivity some operations can usually be performed in parallel.

A note about the sequential nature of the gates has been added to the text on page 5:

“While we have highlighted the advantages of this processor in terms of random access and minimal control hardware, a resulting requirement is the need to perform sequential operations. The number of modes which can be multiplexed to a single qubit without loss of fidelity, is given by the ratio of the loss from idling in a cavity mode to the loss in performing qubit operations, which for modern 3D cavities can be up to 100.”

- It is unfortunate that the resonator T1 times are actually smaller than the transmon’s T1. If the authors have an idea of the origin (the coupling capacitors?), it would be interesting to mention. The advantage of the resonators remains the increased T2*.

Our leading hypothesis for the origin of the lower than state-of-the-art planar resonator coherence times is, as the reviewer mentioned, the use of interdigitated capacitors as couplers. These capacitors, while allowing for large couplings with small footprints, generally have large surface participation, and as a result, are more susceptible to loss via defects on the substrate, such as two-level systems (TLSs).

- I have a slight objection to label the normalized read-out signal as a transmon excitation probability between 0 and 1 (e.g. in Fig. 2bc), especially with a read-out fidelity between 0.3 and 0.85. I guess this also causes the error bars in Fig. 3b to go above 1, which is a little odd. Although it does not affect the story at all, it would be worth a sentence or two in the main text.

The readout fidelity we mention in the supplement is the single-shot readout fidelity, which describes how well the state can be determined in a **single shot**. However, the readout signal can be calibrated to give very high visibility ($\sim 99\%$). This is consistent with our RB data which shows $>98\%$ fidelity of single qubit gates. As part of the investigation we realized that our RB data was taken at an elevated temperature (60mK) resulting in a small amount of leakage from the qubit subspace, leading to an apparent $>100\%$ population. We have replaced the qubit RB data with one from the latest cooldown of the sample, where the qubit temperature was as specified in the supplementary information (20 mK). We have added a line in the supplementary information regarding the readout signal calibration.

"The readout signal is calibrated by appending a sequence with no pulse, and one with a transmon $|g\rangle - |e\rangle \pi$ pulse at the of each set of experimental sequences. Upon averaging over 1000-2000 experiments, the readout signal results in a visibility of $\sim 99\%$, limited by the fidelity of the single qubit gates, and consistent with the RB data."

II. REFEREE 2

This work describes the implementation of a quantum processor comprising an 11-quantum-bit memory, realized as a coupled cavity array of superconducting resonators, and a single processing unit, a transmon-type structure, located at the edge of the array and yet coupled to all its eigenmodes. Coherent interaction between the transmon and a target eigenmode is enabled on-demand by parametric modulation of the transmon frequency. A modification of this scheme enables controlled interactions between any two eigenmodes, which, together with standard single-qubits gates, realize a universal set of pair operations within the quantum memory. The scheme also allows for the measurement of arbitrary multi-bit correlators with minimal overhead.

This is a solid, well-written, and carefully documented piece of work. Connectivity is **the** issue in current efforts aimed at quantum computing, and the multimode approach demonstrated here has some advantage over, say, nearest-neighbor interactions, which is evident in the scaling of two-qubit gate fidelities with the distance between the sites (on the other hand, a disadvantage of a single processing unit, if I understand it correctly, is that pairwise operations need to be implemented sequentially). While frequency crowding will likely limit the number of eigenmodes (and the speed of the gates), the modularity of the scheme makes it worth some consideration for near-term quantum computation and quantum simulation efforts (one could put, for instance, another transmon at the end of the array, and have these transmons shared between more arrays). Qubit lifetimes, resonator quality factors, and gate fidelities reported in this work are somewhat below the state of the art, but I believe that this is not due to an intrinsic limitation in the scheme and, therefore, should not diminish the importance of this work for the community. Before I make my final recommendation, I would like to ask the authors to carefully consider the technical comments below.

1. How do the coherence times of the eigenmodes compare to the quality factor of an individual resonator (fabricated and measured in a similar way)? Is there any relation between the spatial distribution of the modes and their coherence time? In the same way as you do "Hamiltonian reconstruction", could you do "loss reconstruction" and locate where the losses are in the array?

We have measured individual resonators with similar processing and have found longer lifetimes ($Q \sim 10^6$). We suspect that the loss in the eigenmodes is a result of the interdigitated capacitors used to couple the resonators in the chain, with their larger surface to volume participation ratio allowing for enhanced coupling to two-level systems in surface substrate. In principle, the loss can be reconstructed as well, and this would be an interesting direction to pursue. However, we did not do this here as the uncertainty in this calculation tends to be a bit higher, since the loss is small compared to the resonator frequency.

2. You find that T_2^* of the eigenmodes are not T_1 -limited, what is you explanation?

Some of the mode T_2^* s are indeed not T_1 limited. We do not yet know the exact cause for this. We have considered frequency fluctuations inherited by the modes as a result of coupling to the transmon, but this implies lower T_2^* for modes at the center of the band with stronger coupling to the transmon, inconsistent with the observed trends.

3. You have chosen to terminate the array with capacitors of the same size as the inter-site capacitors, so that (for otherwise identical resonators) the edge sites have the same bare frequency as any other and the Hamiltonian is of the form (1). That's fine (I would still mention it), but I find it misleading that in Fig. 1a you denote with g_r the capacitor that goes to ground. That capacitor does not correspond to an coupling element in the Hamiltonian, it just contributes to the total capacitance of the site. Furthermore (if I understand it correctly), the only reason why $g_q \neq g_r$ is that the transmon (as an anharmonic resonator) has a different impedance than the other sites. I would clarify these points and, in general, not associate g_r, g_q with the capacitors in Fig. 1a.

The labels on the capacitors have been removed to avoid direct association of the capacitors with the coupling constants.

4. You find that modes 2 and 8 are much more weakly coupled than the others to the transmon (but still addressable). From "Hamiltonian reconstruction" you obtain weak tunnel coupling between sites 7 and 8 and 9-10, which you ascribe to two capacitors being "defective". Does it mean that their actual capacitance is much smaller than design value? But if it were so, should we not expect the bare frequencies of sites 7-10 to be all higher than the other sites? Please explain.

The defects we mention are in the coupling capacitors, we have added the word "coupling" in the text of the paper. A reduced value of the coupling capacitance does affect the resonator frequency, but not as strongly as the capacitor to ground.

5. Page 4, last par., you say "Many of these errors can be reduced or eliminated by coupling a flux-insensitive transmon to the multimode memory". (i) "many of these errors" sounds too generic to me, can you just tell precisely which errors your suggestion is addressing? (ii) by "flux-insensitive" you mean, in this context, "frequency-fixed"?

(i) The errors we are referring to are:

1. Errors due to dephasing of the transmon. This error can be reduced by using a fixed-frequency transmon (thus reducing sensitivity to flux-noise) and enabling parametric operations between the transmon and modes via a flux-tunable coupler.
2. Errors due to spurious dispersive coupling due to the high bare coupling rate between the transmon and modes. The tunable coupler also reduces this error, as the bare coupling rate between transmon and a particular mode can be chosen to be small while that mode is not being addressed.

We have modified the text to specify these errors.

(ii) by "flux-insensitive" we mean "frequency-fixed". This phrasing has been modified in the text to avoid confusion.

Here is the modified text for this section:

"The error from the dephasing can be reduced by coupling a fixed-frequency transmon to the multimode memory using a tunable coupler. Additionally, biasing the tunable coupler at a point with small static coupling also reduces coherent errors from the the bare dispersive shift."

6. Is it "impossible" in your scheme to enable multiple pair interaction at the same time? Or it can be done under certain conditions? I imagine that feature may find use in some applications.

It is possible to perform linear couplings (such as beam splitter interactions between modes) simultaneously. It is also possible to perform simultaneous interactions between the transmon and multiple modes using multiple simultaneous sidebands. However, there still remains a similar non-linear coupling rate limit. The referee's speculation is an area of interest for us but given the subtlety of the discussion we felt that it was not studied in this work sufficiently to explore in the manuscript.

7. In the Suppl Inf, Sec.V F, is the text accompanying Fig. 9 missing?

Section V. F. contained only Fig. 9, as the figure and caption provide all the information about the coherence measurements of the modes. Some of the text from the caption has been moved to the section text to avoid confusion.

III. REFEREE 3

In the manuscript Random access quantum information processors, the authors perform a proof-of-principle experiment towards an ideal processor showing random access, i.e. the capacity of the processor to directly access to every qubit in the quantum memory. They consider a superconducting transmon qubit playing the role of a central processor and the collective modes of an array of resonators as memory. The selective coupling between the processor and the eigenmodes of the array is performed via stimulated vacuum Rabi oscillations, which allows a faster state transfer than the usual resonant vacuum Rabi oscillations, which allow them to perform a universal set of single- and two-qubit gates and prepare highly entangled states, such as GHZ and Bell states. The authors claim that the approach is scalable and makes use only of the state-of-the-art technology.

In the battlefield which means achieving quantum supremacy in quantum technologies, and especially in superconducting circuits/circuit QED, there are roughly speaking two main approaches. The first one is led by Prof. Schoelkopf in Yale, among others, which privileges the large coherence times of resonators, using qubits as mere non-linear elements to perform operations, and codify the information in cat states. The second one, led by Prof. Martinis at UCSB/Google and Prof. DiCarlo in Delft, for instance, prefers to codify the information in superconducting qubits due to the possibility of direct manipulation, consigning resonators to perform measurements. In this context, the approach followed by the authors, although closer to the first approach, is original since it codifies the information in collective modes and looks for direct control of each memory qubit by the processor.

However, there are claims in this manuscript which are clearly oversold. For instance, they claim that their approach is scalable and it is not in a straightforward manner. Just by taking Eq. (3) of the Supplementary Material, one may see that the difference between consecutive eigenenergies is upper-bounded, in the worst case, by $\Delta E \leq 4g_r \frac{\pi}{n+1}$. This means that, for the current case $n = 11$, they are already upperbounded by a separation of g_r (in fact, following their data, it is even worse, since $\Delta E \approx 50 - 150$ MHz, while $g_r \approx 250$ MHz. My impression is that they are close to the limit, since for larger n , the management of the Landau-Zener transitions should be cumbersome. A possible solution could be a kind of distributed memory in modules, each of them controlled by a single qubit, which can communicate. Indeed, the authors already foresee the problem and propose the aforementioned solution. However, as in the classical case, it will lead to problems such as von Neumann bottlenecks.

The referee's comments about challenges in scalability are astute. In fact all quantum computing technologies have extreme challenges in order to perform large scale error corrected quantum computation. This architecture is no exception. Specifically the referee mentions the frequency crowding limit, which is true for all frequency multiplexed addressing schemes. As the referee states, for a given total bandwidth, B , the maximum rate of access to one of n modes is limited to B/n . We do note that our parametric addressing scheme mitigates the need for traversing Landau-Zener crossings. As mentioned by the referee and the paper, scaling further will require a modular approach. While any multiplexed architecture will have bottlenecks, we also note that if mode coherence is significantly better than qubits (3D resonators have 100x coherence times of the best qubits), adding more memory modes does not introduce significant error despite the multiplexing.

As a consequence of my previous objection, I am also skeptical about the statement that it is a promising route to achieve quantum supremacy. An optimistic estimation of the resources for the implementation of Grover's algorithm, the simplest useful quantum algorithm, with error correction makes use of the order of 500 million fully-controllable qubits. Consequently, this claim is respectfully like stating that jumping is a promising route to reach Alpha Centauri. Moreover, the experiment is interesting by itself and I do not think that it needs this kind of asserts.

We enjoy the referee's colorful analogy. For all quantum computing researchers' sakes we hope that quantum computing is more like reaching Mars or Europa than Alpha Centauri, so that we may live to see it happen. We do dispute the assertion that no quantum advantage can be attained with small numbers of qubits (~ 50) and relatively low-depth circuits which do not require error correction. Our assertion that this architecture can show a quantum advantage, refers to the paper "Characterizing Quantum Supremacy in Near-Term Devices" recently from UCSB/Google, where they argue that: "quantum supremacy can be achieved in the near-term with approximately fifty superconducting qubits" (quote). While in its infancy there appear to be promising applications in quantum chemistry as well as for simulating fundamental condensed matter physics. We believe that with 3D cavities tens of modes could be supported with (multiplexed) fidelities comparable to the best 2D qubits, though we will have to do the next experiments to find out for sure.

I also do not agree with the easy claim that the methods described in this work are compatible with the use of 3D cavities. It is true that the coherent times of 3D cavities are around tens of milliseconds, but it does not directly mean that one may use the same approach, since the energy gaps between

the collective modes could be smaller due to the degeneracy introduced by the orthogonal modes (see for instance, the works performed in the group of Prof. Gross at the WMI, Garching), which could destroy the information codification.

The referee has brought up one of the key issues in adapting the presented architecture to 3D cavities. There are several geometries that can give appropriate dispersions of modes, the simplest of which is a rectangular waveguide cavity with a single long dimension that gives TE_{10n} modes with no degeneracies.

Even though the experiment is a proof of principle, quantum technologies and especially superconducting circuits / cQED are at the level of producing already applications beyond a set of universal quantum gates. Additionally, Nature Communications is an interdisciplinary journal, in which the articles should be accessible to non-specialist readers. In this sense, from my point of view, it is not only useful, but fundamental, to frame the new results in the state of the art. For instance, to my knowledge, the most advanced digital quantum simulation in this platform is achieved in R. Barends et al. Nature 534, 222 (2016). A fair comparison of the gates in terms of fidelities, times, robustness, etc is important, does not diminish the achievements of the authors, and could provide a more objective point of view to a non-specialist reader.

We agree with the referee, that this would provide useful context. We have updated the benchmark line in Fig. 4 to use the fidelity set by Barends, et. al. and have included a reference in the text.

"Even without considering potential improvements in the coherence times, we see (Figure 4c) that the processor performs competitively with state-of-the-art gates [34] between distant qubits in a nearest-neighbor architecture."

Additionally, the authors can find few more brief complementary comments about their manuscript.

A. Other Comments

1. I do not understand why the authors use the terminology memory bits and qubits alternatively. Sentences such as The memory bits are superpositions of vacuum and single-photon states in the abstract are conceptually wrong, since bit is a measurement of classical information. If the authors do not want to use the word qubit, the alternative is quantum bit. Otherwise, it seems that the memory is classical and only able to store classical information.

The text has been edited to universally state that the memory is quantum memory and use of the term bit has been replaced with the word qubit.

2. Interdigitated capacitors is used only in the caption of Fig. (1). As this is not a broadly used terminology, I suggest them to replace it by capacitively coupled strongly to each other.

The caption has been edited as suggested.

3. The points in Fig. (4c) are mainly indistinguishable. I suggest them to put the origin in a fidelity of 50%, which is essentially the worst possible fidelity for a quantum gate (corresponds to make it randomly), writing the legend in two columns, for instance.

The origin of Figure 4c has been changed to make the data points more distinguishable, as per the referee's suggestion.

4. With respect to the figures in the Supplementary Material, Fig. (5) has no legend in a., b. and d. Figures (7), (8), (9a), (10b) and (13) have no error bars.

We have added legends to Fig. (5) and have added error bars to the other figures. Explanations of how the error bars are extracted have been included in the figure captions.

5. As far as I see from Section IX-B of the Supplementary, the fidelity of Bell states is around 75%, which is compatible with the fidelity of the SWAP gate. However, the authors claim that they can construct GHZ states by repeating the swapping multiple times. With $n = 11$, it means the same number of entangling gates. As a consequence, I would expect that the final GHZ state would show a terrible fidelity. It seems compatible with the fact that in Fig. (5c) mode occupation error percentage is provided, but no the fidelity. Authors must provide this value and, if it is too bad, they should avoid the claim that they can construct this state.

The purpose of Fig. 5c is to illustrate how such states can be created. The demonstrated protocol is quite efficient, requiring only one transmon-mode swap for each bit in the GHZ. Therefore, the error per added mode scales closer to the single-mode benchmarking data (89 – 97% fidelity). We have checked the fidelity of the Bell state (75 – 80%). As is evident in the Bell state tomography and the process tomography shown in Fig. 4, much of the error is actually in the tomography. The reason tomography was not performed on higher states is that the tomography process would dominate the error. However, we have measured the Mermin witness operator for 3-qubit GHZ, demonstrating that the state is entangled and within the GHZ class. We have added a section to the supplemental information showing the 3-qubit entanglement witness.

We have modified the main text to qualify the entanglement claim as follows:

“While the three mode GHZ state can be demonstrated to be tripartite entangled through a measurement of the Mermin witness (Supplementary Information), full characterization of entangled states of more than two modes is hampered by the additional gates required for tomography and the gate fidelities of the current device. This protocol however illustrates the ease with which a random access quantum information processor can be used to generate multimode entangled states of arbitrary modes.”

6. Finally, in References [9], [17] and [19] of the Supplementary Material, some capital letters are missing.

These references have been corrected to the appropriate capitalization.

REVIEWERS' COMMENTS:

Reviewer #2 (Remarks to the Author):

I am satisfied with the authors' response to my comments. Some aspects of their work are worth further attention (T_2^* of resonators, gate parallelization, full characterization of GHZ states), but the results presented suffice for a proof-of-principle demonstration.

In their reply, the authors have also addressed Referee 3's objections to the scalability of the technique (limited by frequency crowding), and to the possibility of achieving quantum supremacy in the short/mid-term. These objections could as well be directed to a large share of recent work on quantum information processing with superconducting circuits; indeed, these are crucial and open questions at the heart of the field. Clearly, the present approach does not give a definite answer to these questions. Still, it indicates a direction that looks competitive with respect to existing approaches, and therefore worth pursuing. After all, at the present stage this race has no clear winner, yet.

For these reasons, I recommend publishing this work in Nature Communications as it is.

Reviewer #3 (Remarks to the Author):

We do dispute the assertion that no quantum advantage can be attained with small numbers of qubits (≈ 50) and relatively low-depth circuits which do not require error correction. Our assertion that this architecture can show a quantum advantage, refers to the paper "Characterizing Quantum Supremacy in Near-Term Devices" recently from UCSB/Google, where they argue that: "quantum supremacy can be achieved in the near-term with approximately fifty superconducting qubits" (quote). While in its infancy there appear to be promising applications in quantum chemistry as well as for simulating fundamental condensed matter physics. We believe that with 3D cavities tens of modes could be supported with (multiplexed) fidelities comparable to the best 2D qubits, though we will have to do the next experiments to find out for sure.

I am sorry, but I am not into using *argumenta ad verecundiam* (appeal to authority arguments) in science. In order to achieve quantum supremacy with a purely digital approach and quantum correction, millions of qubits are required, or in other word, in this moment there is no a single example of digital protocol with error correction achieving quantum supremacy with a few qubits. For reaching quantum supremacy with a small number of qubits, a digital-analog approach is required, and examples are the quantum speckle from Google, spin glasses, boson sampling, etc.

Furthermore, in the sentence *While in its infancy there appear to be promising applications in quantum chemistry as well as for simulating fundamental condensed matter physics*, I am not going to embarrass you asking about a precise model and algorithm of such promising applications, because there is no a single example until now of a useful quantum supremacy problem with few qubits. Even though I suggest the authors to remove this claim for the sake of honesty, I understand that we are living times of overselling and press highlights.

In any case, I consider that the authors have all in all addressed my main complaints and hence, the manuscript is, from my point of view, suitable for Nature Communications.

I. REFEREE 2

I am satisfied with the authors' response to my comments. Some aspects of their work are worth further attention (T_2^* of resonators, gate parallelization, full characterization of GHZ states), but the results presented suffice for a proof-of-principle demonstration. In their reply, the authors have also addressed Referee 3's objections to the scalability of the technique (limited by frequency crowding), and to the possibility of achieving quantum supremacy in the short/mid-term. These objections could as well be directed to a large share of recent work on quantum information processing with superconducting circuits; indeed, these are crucial and open questions at the heart of the field. Clearly, the present approach does not give a definite answer to these questions. Still, it indicates a direction that looks competitive with respect to existing approaches, and therefore worth pursuing. After all, at the present stage this race has no clear winner, yet. For these reasons, I recommend publishing this work in Nature Communications as it is.

We are happy that the referee is satisfied with our responses to his questions and recommends publication.

II. REFEREE 3

I am sorry, but I am not into using *argumenta ad verecundiam* (appeal to authority arguments) in science. In order to achieve quantum supremacy with a purely digital approach and quantum correction, millions of qubits are required, or in other words, in this moment there is no single example of digital protocol with error correction achieving quantum supremacy with a few qubits. For reaching quantum supremacy with a small number of qubits, a digital-analog approach is required, and examples are the quantum speckle from Google, spin glasses, boson sampling, etc. Furthermore, in the sentence *While in its infancy there appear to be promising applications in quantum chemistry as well as for simulating fundamental condensed matter physics*, I am not going to embarrass you asking about a precise model and algorithm of such promising applications, because there is no single example until now of a useful quantum supremacy problem with few qubits. Even though I suggest the authors to remove this claim for the sake of honesty, I understand that we are living times of overselling and press highlights. In any case, I consider that the authors have all in all addressed my main complaints and hence, the manuscript is, from my point of view, suitable for Nature Communications.

While we disagree with the referee regarding the quantum advantage that can potentially be achieved with tens of qubits, we understand the referee's position. We have removed the objectionable sentence from the manuscript.

We believe we have addressed all the concerns raised and look forward to the publication of our paper in Nature Communications.

Sincerely,
Ravi Naik and David Schuster